# Polycyclic aromatic hydrocarbon formation chemistry in a plasma jet revealed by IR-UV action spectroscopy

Alexander K. Lemmens[1,2], Daniël B. Rap[1], Johannes M.M. Thunnissen[1], Bryan Willemsen[1] & Anouk M. Rijs [1]*

Large polycyclic aromatic hydrocarbons (PAHs) are the most abundant complex molecules in the interstellar medium; however, their possible formation pathways from small molecular species are still elusive. In the present work, we follow and characterize the formation of PAHs in an electrical discharge, specifically the PAH naphthalene in a molecular beam of argon. The fragments, products and reaction intermediates are unambiguously structurally identified by mass-selective IR-UV spectroscopy combined with quantum chemical calculations. This experiment provides evidence of the formation of larger PAHs containing up to four cyclic rings in the gas phase originating from a non-radical PAH molecule as a precursor. In addition to PAH formation, key resonance stabilized radical intermediates and intermediates containing di-acetylenic side groups are unambiguously identified in our experiment. We thereby not only reveal competing formation pathways to larger PAHs, but also identify intermediate species to PAH formation that are candidates for detection in radio-astronomy.

[1] Radboud University, Institute of Molecules and Materials, FELIX Laboratory, Toernooiveld 7c, 6525 ED Nijmegen, The Netherlands. [2] Van 't Hoff Institute for Molecular Sciences, University of Amsterdam, Science Park 904, 1098 XH Amsterdam, The Netherlands. *email: a.rijs@science.ru.nl

Polycyclic aromatic hydrocarbons (PAHs) are considered ubiquitous carriers of carbonaceous material in the interstellar medium (ISM). They are responsible for the infrared emission features called the aromatic infrared bands[1] in the 3–20 μm region and considered promising candidates for the diffuse interstellar bands[2] that are present between about 400 and 1200 nm. In some astrophysical objects up to about 20% of carbon is estimated to be locked up by PAHs with a typical size of more than 50 carbon atoms[3]. PAHs also participate in the complex network of reactions that occurs in stellar ejecta as constituent or even catalysts, thereby affecting the overall chemical composition of star- and planet-forming regions[3,4]. If the chemistry of PAHs in relevant environments is thoroughly understood, then the PAH molecules could also act as a probe for the physical environment that surrounds them.

The exact formation mechanisms and chemistry of large PAHs in the ISM, however, are still under debate and ask for laboratory experiments under controlled conditions[3–5]. Currently, it is especially of interest to elucidate these formation mechanisms now that both large aromatic systems and a small aromatic molecule have been identified by astronomical observations in combination with laboratory spectroscopy[6–8] and it is expected that more molecule identifications will follow.

Bottom-up and top-down gas phase reaction mechanisms that lead to the formation of PAHs are both considered[1] and as of yet, the formation of larger PAHs from smaller PAHs has not been thoroughly explored in controlled conditions in a laboratory. Therefore, exact intermediates and possible reaction steps are still unknown. The current knowledge on PAH growth is mostly based on combustion and flame experiments, where PAH growth is explained mainly by the hydrogen abstraction–acetylene addition (HACA) mechanism. However, these studies start from reactive precursors at elevated temperatures[9] and are limited to the formation of bi- and tricyclic PAHs[10–13]. In the HACA mechanism described by Frenklach et al.[14] PAHs grow in a multi-step reaction that involves acetylene. First, an aromatic radical reacts with an acetylene radical, which is followed by abstraction of molecular hydrogen. Subsequently, a second acetylene molecule is added to the ring structure followed by ring closure. Apart from the HACA mechanism, PAH growth reactions including vinylacetylene are also found to occur with extremely low activation barrier, illustrating the efficiency of two-body instead of multi-body reactions[10]. While the HACA mechanism has remained central in explaining the formation of small PAHs, both the necessary complementary reaction pathways to tricyclic PAHs and larger PAHs created under controlled conditions are still elusive[11–13,15–18]. In relation to astrochemistry, plasma sources have often been used to create a suitable chemically reactive environment, i.e. gas phase and cold[19–25]. For example, it was shown that polyyne and small (transient) cyclic compounds can be formed in a discharge source. These were identified using high-resolution techniques such as cavity ring-down or microwave spectroscopy and still belong to the largest carbon species detected in the ISM with the exception of fullerenes[26].

In stellar outflows or the ISM energetic particles and photons ionize atoms and molecules, thereby inducing gas phase synthesis of larger molecules[27,28]. In our experiment, we perform ionization/radicalization and subsequently induce gas phase reactions by means of an electrical discharge. Although conditions are not analogs, reaction routes that occur in this discharge are of interest for astrochemistry as possible growth pathways for PAHs. Much is still unclear on the growth mechanisms of PAHs in these plasma environments, mainly because one has had to rely solely on mass spectrometry[29–32]. Here, we identify reaction products and intermediates through mass-selective IR-UV ion dip spectroscopy and study the reactions of PAH growth at a deeper level. This versatile approach enables us to unambiguously assign reaction products using the obtained mass-selective IR-fingerprint spectra in combination with quantum chemical calculations[33–35]. Together with the reaction products, we also identify key intermediate species that are crucial for understanding interstellar chemistry. Not only does the observation of intermediates in the reaction process aid in elucidation of the main mechanism, it also provides chemical species with a dipole moment suited for astronomical searches in radio-astronomy. This region of the electromagnetic spectrum complements the infrared regime and allows for direct comparison with telescope observations[36,37].

## Results

**Detection of products**. Figure 1a displays a typical mass spectrum of the discharge of naphthalene probed via $1 + 1'$ REMPI spectroscopy, where the first photon is resonant with an unresolved vibronic band of an electronic state of the formed molecules. The parent signal ($m/z = 128$) is truncated to visualize the lower intensity signals. Fragmentation of the parent (naphthalene) results in fragments with $m/z < 128$, whereas reaction products are observed with $m/z > 128$. The REMPI scheme is required to probe

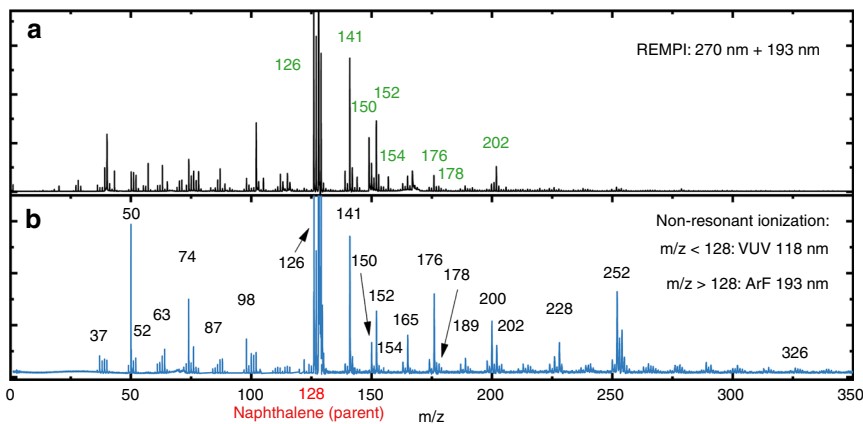

**Fig. 1 Mass spectra of products and fragments from naphthalene formed in electrical discharge.** Trace **a** is recorded with two-color REMPI during our IR-UV ion dip measurements. Products that are assigned to a molecular structure are labeled with their masses in green, see Figs. 2 and 3. Trace **b** is taken using 118 nm VUV for the region below $m/z = 128$ and 193 nm UV for the region above $m/z = 128$. The $m/z = 326$ would correspond to a formed cyclic structure containing seven aromatic rings.

IR absorption which leads to depopulation of molecules that are prepared in the ground state, thereby recording mass-selective IR-UV ion dip spectra[38]. The high-intensity mass channels in the 1 + 1′ REMPI mass spectrum are probed using the IR-UV ion dip technique. Note that the observed intensity does not directly reflect relative abundance since excitation and ionization cross-sections vary between different species. Figure 1b shows the mass spectrum recorded via non-resonant ionization with either 193 nm (produced with ArF excimer laser) or ionization with higher energy photons of 118 nm VUV (produced using an Xe HHG cell). In general, larger molecules are ionized more readily using 193 nm UV light while smaller species are more prone to ionization by the 118 nm VUV light[39]. By monitoring relative ion signals of products, photodissociation using the non-resonant ionization techniques is avoided. The non-resonant ionization allows for the ionization of species with up to $m/z = 326$. Doubly ionized species are not expected with photoionization, thus the $m/z$ value reflects the mass of species created. In addition to the photoionization mass spectra we have obtained a mass spectrum of the cations produced in the electrical discharge (see Supplementary Fig. 1). The cationic mass spectrum provides a good reference for relative abundancies of the different products that are produced with no chance of photofragmentation.

The lower mass fragments from naphthalene ($m/z < 128$) after discharge consist mostly of carbon atoms with only a few hydrogen atoms. Similar masses are previously observed by Linnartz and co-workers over the years[21–23,40,41]. Typical examples of formed fragments and low mass products are $C_3$ ($m/z = 36$), $C_4H_2$ ($m/z = 50$), $C_5H$ ($m/z = 61$), $C_6H_2$ ($m/z = 74$), and $C_8H_2$ ($m/z = 98$). These result typically from the discharge of a mixture containing acetylene ($C_2H_2$, $m/z = 26$) and argon. However, the main neutral small fragment that is produced in our experiment is observed at $m/z = 50$, which most probably corresponds to diacetylene ($C_4H_2$). As will be shown in this work, the formation of diacetylene seems to be a key ingredient and an important contributor to PAH formation. This is in contrast to what was previously considered, as the current HACA mechanism describes the addition of mono-acetylene units. Diacetylene is already proposed to be a potential competitor for radical pathways to form large hydrocarbons[42] and $C_4H_2$ has been detected in Titan's atmosphere[43] and the ISM[7]. For the molecules with masses larger than the parent molecule naphthalene ($m/z > 128$), only the chemical composition may be determined from the $m/z$ values. The chemical structures cannot be deduced from the $m/z$ value alone and spectroscopic information is required to ascertain the molecular formulae and/or structures. For example, the addition of two mono-acetylene functional groups to the precursor naphthalene would result in the same mass as if one diacetylene unit would be attached. In a similar fashion, the formation of acenaphthylene cannot be distinguished from ethynylnaphthalene as both reaction products would have an $m/z$ of 152.

**Structure identification of substituted PAHs.** Where the chemical formula of small species can be deduced from their mass, spectroscopic experiments are necessary in order to elucidate the chemical formulae and structures of the products of naphthalene ($m/z > 128$). Using non-resonant ionization we are able to observe PAH formation up to $m/z = 326$, which would correspond to PAHs containing seven cyclic rings. In order to assign the molecular structures of the masses indicated in green in Fig. 1a, we rely on the combination of mass-selective IR spectroscopy in the fingerprint region (600–1800 cm$^{-1}$) and quantum chemical calculations to assign the experimental IR spectra[44].

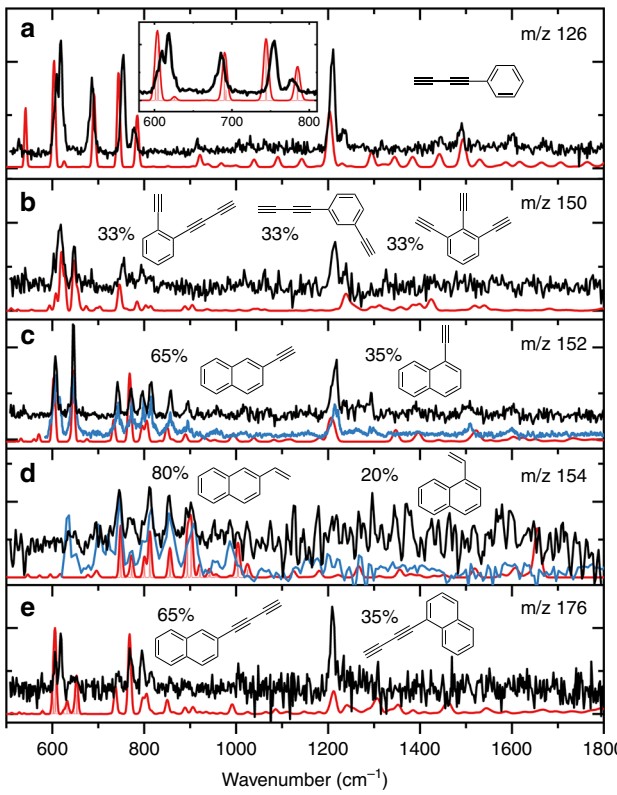

**Fig. 2 Mass selected IR spectra (black) of discharge products with the assigned theoretical IR spectra (red).** The mass channels shown are $m/z=$ **a** 126, **b** 150, **c** 152, **d** 154, and **e** 176 and are recorded using 36616 cm$^{-1}$ UV photon energy. The molecules assigned are **a** buta-1,3-diyn-1-ylbenzene (diacetylenebenzene); **b** 1-(buta-1,3-diyn-1-yl)-2-ethynylbenzene, 1-(buta-1,3-diyn-1-yl)-3-ethynylbenzene, and 1,2,3-triethynylbenzene; **c** 1- and 2-ethynylnaphthalene (1EN and 2EN); **d** 1- and 2-ethenylnaphthalene; and **e** 1- and 2-(buta-1,3-diyn-1-yl)naphthalene. For **b**–**e**, two or more theoretical IR spectra of structural isomers are used to fit the experiment allowing for the determination of the ratio between the formed isomers (individual calculated spectra can be found in Supplementary Figs. 2–5). The blue trace in the $m/z$ 152 box corresponds to an FT-IR reference spectrum taken from 1EN and 2EN combined with a 1:2 ratio, respectively. The blue trace in the $m/z$ 154 box corresponds to an FT-IR reference spectrum taken from 1- and 2-vinylnaphthalene combined with a 1:4 ratio, respectively. The light red traces correspond to stick spectra to display adjacent peaks.

By comparison of theoretically calculated IR spectra of multiple possible isomers to experimental spectra we are able to assign the reaction products with $m/z = 126$ to (a) buta-1,3-diyn-1-ylbenzene (diacetylenebenzene); $m/z = 150$ to (b) 1-(buta-1,3-diyn-1-yl)-2-ethynylbenzene, 1-(buta-1,3-diyn-1-yl)-3-ethynylbenzene, and 1,2,3-triethynylbenzene; $m/z = 152$ to (c) 1- and 2-ethynylnaphthalene (1EN and 2EN); $m/z = 154$ to (d) 1- and 2-ethenylnaphthalene; and $m/z = 176$ to (e) 1- and 2-(buta-1,3-diyn-1-yl)naphthalene as displayed in Fig. 2. In order to assign these observed masses, we have calculated a variety of IR spectra originating from possible isomers that coincide with the observed mass, see Supplementary Figs. 2 to 5 for a detailed assignment. The unambiguous structural identification that is described in the Supplementary Figs. 2–7 shows both the accuracy as well as suitability of mass-selective IR spectroscopy in determining discharge reaction products. Certainly as important as assigning reaction products, some species can be discarded based on a mismatch between the experimental and calculated IR spectra. For several product mass channels, multiple constitutional (structural) isomers are computed and their IR spectra are

combined in order to match the observed IR spectra. For example, the calculated IR spectra of 1-ethynylnaphthalene and 2-ethynylnaphthalene (1EN and 2EN respectively) (see Supplementary Fig. 4) are summed in a ratio that will match the intensities observed in the experimental spectra as shown in Fig. 2c. Supplementary Fig. 4 also illustrates that acenaphthylene can be discarded based on the mismatch between the calculated and experimental IR spectrum.

Overall, an excellent agreement between theoretical and experimental IR spectra is found. The IR region between 550 and 1000 cm$^{-1}$ of the vibrational spectrum is most important, since both the highest intensity and most diagnostic bands can be found here. This region is often sufficient to assign a calculated spectrum of a particular molecular structure to the experimental spectrum. The bands around 600 cm$^{-1}$ arise from ≡CH deformation, present in Fig. 2a–c, e. Around 800 cm$^{-1}$ the CH out-of-plane of aromatic rings are found. The small deviations between experimental frequencies and theoretical frequencies as shown in the inset of Fig. 2a below about 700 cm$^{-1}$ are common and usually corrected for by using multiple scaling factors[45]. A notable feature in the remaining part of the experimental IR spectra arises from reaction products that contain a ≡CH group. For these molecules an intense band at 1210 cm$^{-1}$ is observed, which is predicted using anharmonic theory (e.g. in Fig. 2a). This peak is the result of an overtone arising from the ≡CH deformation as is known in literature[46–48] and shown by anharmonic calculations and by gas phase FT-IR spectra of reference compounds by ourselves and Constantinidis et al.[49]. In order to cement the assignment of structures presented in Fig. 2, we have compared the reference gas phase FT-IR spectra of 1EN and 2EN, i.e. the assigned compounds presented in Fig. 2c, with $m/z = 152$ (see Supplementary Fig. 7 as well as the blue trace in Fig. 2c). Excellent agreement between the reference compound FT-IR spectra of 1EN and 2EN and the IR-UV ion dip spectrum of $m/z = 152$ recorded after the discharge of naphthalene is observed for the diagnostic region between 550 and 1000 cm$^{-1}$ as well as the anharmonic band at 1210 cm$^{-1}$. The peak at 1210 cm$^{-1}$ appears in all experimental spectra where, based on the 550–1000 cm$^{-1}$ region, ethynyl functional groups are expected (Fig. 2a–c, e). In contrast, the experimental IR spectrum presented in Fig. 2d assigned to ethenylnaphthalene does not show this peak at 1210 cm$^{-1}$. The intense band at 1210 cm$^{-1}$ can act as a diagnostic tool for the identification of ≡CH containing structures. Moreover, this is a clear demonstration of how anharmonicity influences not only the mid-IR spectra of PAHs[50] but also the IR spectra of PAH substituents and PAH growth intermediates.

**Identification of larger PAHs**. Besides the substituted PAHs, we are also able to identify fully cyclic structures that are larger than the original PAH naphthalene, see Fig. 3. The top panel of Fig. 3 shows the two color $[1 + 1']$ UV excitation spectrum obtained for $m/z = 178$, which can be assigned to phenanthrene by comparing this REMPI UV spectrum to a previously reported spectrum of phenanthrene[51]. Similarly to our IR spectra, high-resolution REMPI spectra also provide a unique fingerprint of a molecule and especially together with the mass of an unknown species, a few transitions are sufficient for spectroscopic assignment. The obtained UV spectrum after discharge is somewhat broader than the literature spectrum[51] due to thermal-broadening; however, the $S_2$ origin as well as the first few vibronic bands are clearly observed. We hereby simultaneously show that we can use UV ion spectroscopy to identify molecules created in a discharge and that larger PAHs are formed in an electrical discharge.

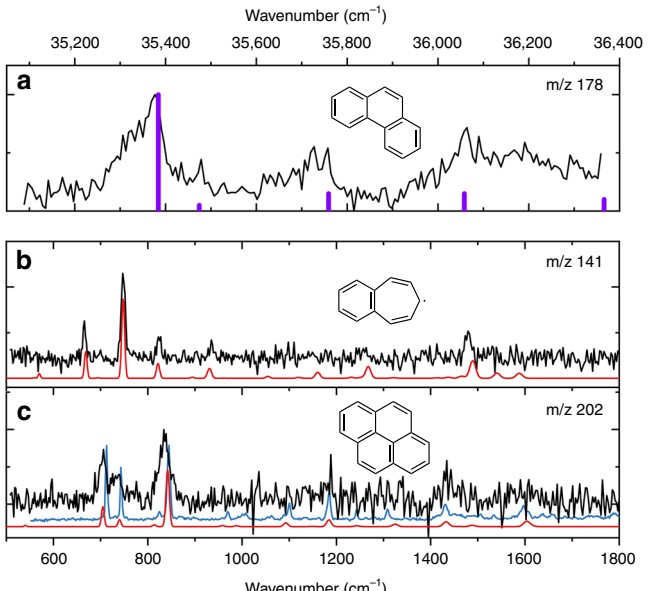

**Fig. 3 Assignment of larger PAHs formed in the discharge using REMPI and IR ion dip spectroscopy. a** Experimental REMPI spectrum of the $m/z = 178$ channel (black) with literature values of the spectrum of phenanthrene [28] (mass 178 amu) as reference (violet). **b**, **c** Experimental IR-UV ion dip spectra recorded using 36616 cm$^{-1}$ UV photon energy (black). Theoretical spectra of the assigned molecular structures displayed in red: benzo[7] annulene radical for $m/z = 141$ and pyrene for $m/z = 202$. The IR-UV ion dip spectrum of reference compound pyrene is in blue and was recorded using the methods described in ref. [69].

Two additional fully cyclic compounds are identified using mass-selective IR-UV ion dip spectroscopy, namely the benzo[7] annulene radical and pyrene (with $m/z = 141$ and 202, respectively), see Fig. 3b, c. For both the assignment of benzo[7] annulene and pyrene, the 550–1000 cm$^{-1}$ IR region is most diagnostic and excellent agreement between theoretical and experimental spectra is observed. Methyl-radical addition to the parent naphthalene leading to species with $m/z = 141$ is excluded based on the comparison between the experimental spectrum and calculated spectra, see Supplementary Fig. 6. The anti-symmetric CH out-of plane mode with a frequency of 670 cm$^{-1}$ is the most diagnostic mode, which is clearly present in the calculated IR spectrum of benzo[7]annulene, but absent in the calculated spectra of methyl-radical containing compounds.

## Discussion
In general, two types of PAHs are formed from the discharge of naphthalene, namely larger PAHs and polyyne substituted PAHs. From the latter, the most remarkable of the identified structures presented in Fig. 2 are the diacetylene side chains. The HACA mechanism described by Frenklach et al.[14] and Visser et al.[52], which is currently considered the main reaction pathway, does involve the addition of acetylenic side chains to PAH but lacks intermediates with diacetylene side groups. Here we show that using only naphthalene as a precursor, PAHs grow via the addition of radical di-acetylenic side chains, see Fig. 4. Based on the observed IR signatures in combination with the calculated IR spectra, single acetylene side group addition was excluded for the $m/z = 126$ and 176 channels as can been seen in Supplementary Figs. 2 and 5. Instead, diacetylene moieties are attached to the PAH ring. The formation mechanism of radical diacetylene is not precisely known, although we propose that it originates from the most abundant fragment, diacetylenebenzene ($m/z = 126$). As

**Fig. 4 Proposed reaction mechanism of the formation of phenanthrene and intermediates. a** Reaction mechanism of the formation and subsequent fragmentation of diacetylenebenzene ($m/z = 126$). **b** The diacetylene that is created in step **a** can further react to form diacetylene–naphthalene ($m/z = 176$) which is a likely precursor for the formation of phenanthrene (**c**).

shown in Fig. 4a, opening of one of the naphthalene rings upon electrical discharge forms buta-1,3-diyn-1-ylbenzene (diacetylenebenzene) ($m/z = 126$), and subsequent dissociation of the diacetylene side group of the naphthalene precursor results in free diacetylene radicals. Radical diacetylene can attach to the naphthalene parent molecule, as shown in Fig. 4b, forming the diacetylene–naphthalene isomeric structures ($m/z = 176$). In the reaction region of our discharge nozzle (see Supplementary Fig. 8), the created radicals have ample opportunity to react with the abundant parent molecules that are not fragmented in the plasma zone. We do expect radical-neutral reaction pathways involving the diacetylene radical, since these are more favorable as shown by calculations.

Besides acetylenic moieties, the addition of a vinyl side group is observed in lower quantities. This corresponds well with the minor reaction pathway to PAHs as described by Brittner and Howard[53]. Butadiynyl-ethynylbenzene, the product with $m/z = 150$, is thought to be the result of ring opening of 1EN and 2EN ($m/z = 152$). This confirms our assignment of $m/z = 152$ to ethynylnapthalene, as ring opening would indeed result in $m/z = 150$. The formation of acenaphthylene ($m/z = 152$) can be excluded (see Supplementary Fig. 4), this is notably different to pyrolysis experiments where acenaphthylene is one of the most ubiquitous combustion effluents[52]. The reaction mechanisms presented here and illustrated in Fig. 4a, b show key intermediates and their reaction pathways to the formation of larger PAHs in a plasma containing only naphthalene as a precursor.

The larger, three-membered ring PAH, phenanthrene, is most likely formed by ring closure of the identified compound diacetylene–naphthalene ($m/z = 176$) after hydrogenation as shown in Fig. 4c. The four-membered ring PAH pyrene ($m/z = 202$), which is identified and characterized as well, is believed to be formed by a similar mechanism from phenanthrene via acetylene addition followed by ring closing as the formation of phenanthrene itself. Although we did not find intermediates (e.g. phenanthrene with acetylene side-chain) in our experiment, pyrolysis experiments do point in this direction[52,54]. Unfortunately, ethynyl phenanthrene (phenanthrene with an acetylene side-chain) has the same $m/z =$ as pyrene itself. However, one would expect to observe the anharmonic peak at 1210 cm$^{-1}$ if

traces of this intermediate species would be still present after supersonic cooling of the discharged mixture, thus on this basis of the absence of the 1210 cm$^{-1}$ peak we exclude the ethynyl phenanthrene species.

The peak at $m/z = 141$ has been unambiguously assigned to the resonance stabilized radical benzo[7]annulene. However, no measurable intermediates are identified in our experiment, hence its formation pathways remain uncertain. Possibly the benzo[7]annulene radical is formed via cyclopropanaphthalene, which is known to be able to react to annulene species[55–57]. Johansson et al.[58] stress the importance of resonance stabilized radicals in the growth process of PAHs and soot particles in combustion. Our observation and characterization of the benzo[7]annulene radical substantiates these conclusions. All the intermediate species presented in Fig. 2 and the benzo[7]annulene have significant dipole moments of up to about 2 Debye and are therefore potential candidates for microwave studies such as the one performed by Lee and co-workers[59] and radio-astronomy searches. Up to now, no RSRs have been observed in the ISM and the benzo[7]annulene radical would a promising candidate[60].

In summary, our experiments identify additional reaction pathways for the interstellar chemistry of PAHs. In particular, the chemistry of one of the smallest PAHs, naphthalene, in an electrical discharge, is elucidated by unambiguously assigning reaction products by mass-selective IR-UV ion dip spectroscopy. Growth of naphthalene is observed resulting in the formation of phenanthrene and pyrene, compact 3- and 4-ring PAHs, respectively. Most of the identified intermediates contain acetylene and diacetylene side chains, thereby providing alternatives to the HACA mechanism by Frenklach in which up to now, the growth of PAHs was described via the addition of mono-acetylenic side chains. Our experiments demonstrate that (radical) diacetylene addition has to be taken into account in PAH growth models. Moreover, the identified intermediates also include a resonance stabilized radical which has only recently been put forward as crucial intermediate to promote low barrier reactions[58]. Finally, the identified substituted PAHs and PAH radicals have significant dipole moments and are ideal candidates for radio-astronomy searches.

## Methods

**Experimental**. All experiments were performed at the FELIX laboratory[61] in the Netherlands using a molecular beam apparatus.[62] A gas pulse consisting of naphthalene in argon (80 °C, 0.25%) is discharged at 0.55 kV and 50 mA before being expanded into a vacuum chamber using the approach of McCarthy et al.[19,20]. In this approach, an isolating spacer of 6 mm is used to confine the plasma directly after discharge in order to increase the number of collisions before the expansion to create a longer reaction time. The subsequent expansion "freezes" the reaction after which the reaction mixture is probed. A current of 50 mA was found to be optimal for the formation of products (see Supplementary Fig. 11). Using the Paschen curve (for $\gamma$ between 0.1 and 0.2 (ref. [63]) which corresponds to a pressure between 0.37 and 0.51 mbar) and the relation between electron energy and pressure from Engel[64], the electron energy is estimated to be 3.15–3.45 eV. The rovibrational cooling was increased by changing the geometry of the discharge nozzle with respect to the design of McCarthy as shown in the Supplementary Figs. 8–10: a compression after the discharge but before expansion in vacuum ensures a colder molecular beam (SCOOTER nozzle). The formed fragments, intermediates, and products in the molecular beam are probed using either $[1 + 1']$ resonant enhanced multiphoton ionization (REMPI) at 270 nm or non-resonant ionization by an ArF laser (193 nm) or via direct ionization using the VUV (118 nm) source[39]. Infrared spectra are recorded with IR-UV ion dip spectroscopy using the free electron laser FELIX. Ionized species are detected using a reflectron time-of-flight mass spectrometer equipped with a multichannel plate ion detector.

**Theoretical**. Initial computations for assignment are performed using the Gaussian 16 program[65] on the B3LYP/6-31+G* level of theory[66,67]. Second, in order to simulate the anharmonic 1210 $cm^{-1}$ band in spectra of the molecules in Fig. 2 the option anharmonic is used with the Jun-cc-pVDZ basis set[68] (i.e. for the spectra of buta-1,3-diyn-1-ylbenzene; 1-(buta-1,3-diyn-1-yl)-2-ethynylbenzene, 1-(buta-1,3-diyn-1-yl)-3-ethynylbenzene, and 1,2,3-triethynylbenzene; 1- and 2-ethynylnaphthalene and 1- and 2-(buta-1,3-diyn-1-yl)naphthalene). However, since the perturbation treatment leads to large and unwanted shifts in the fundamental frequencies of bands especially between 500 and 1000 $cm^{-1}$, the derivatives with respect to these normal modes are selectively removed. All IR spectra are scaled with an appropriate scaling factor (0.96 for the Jun-cc-pVDZ and 0.976 for 6–31+G*) and convoluted with 1% of the photon energy to match the FELIX bandwidth.

## Data availability

The x,y-data of the experimental and theoretical spectra and the coordinates of the calculated structures are included in the Supporting Data. Additional data that support the findings of this study are available from the corresponding author upon reasonable request.

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

## Acknowledgements
We would like to thank the FELIX laboratory team for their experimental assistance and scientific support and we acknowledge the Nederlandse Organisatie voor Wetenschappelijk Onderzoek (NWO) for the support of the FELIX Laboratory and SURFsara (proj. 17603) for their computational resources. Furthermore, we would like to thank Prof. Dr. Harold Linnartz and Dr. Lex van der Meer for their useful discussions. At last, we thank Dr. Amanda Steber, Dr. Sébastien Gruet, and Prof. Dr. Melanie Schnell for their cooperation on developing the discharge source.

## Author contributions
A.K.L., D.B.R., J.M.M.T., B.W. and A.M.R. conceived and designed the experiments. A.K.L., D.B.R. and J.M.M.T. performed the experiments and computations. The article was written by A.K.L. and A.M.R.

## Competing interests
The authors declare no competing interests.

## Additional information

7