## [Peer Review File · Nature Communications]

Reviewers' comments:

Reviewer #1 (Remarks to the Author):

The manuscript by Lemmens et al. is a well-written and well-reasoned article that describes the spectroscopic analysis using mass-selective IR and UV techniques when naphthalene is subjected to an electrical discharge. Theoretical calculations are used to aid the identification of various fragments and larger species; I found this portion of the analysis fairly convincing. Key findings include the prominence of intermediate species with C₄H side chains and larger PAHs with 3 and even 4 membered rings. Although the present findings may provide insight into the pathways by which larger PAH molecules are synthesized, it would have been helpful if the authors had demonstrated the same basic findings for a second small PAH. Perhaps for obvious reasons, this experiment might be beyond the scope of the present work, but when and where appropriate, the authors should discuss the generality of their present results. Is there any reason to believe that other small PAHs might behave differently?

It is also worth emphasizing that although the present experiments may indeed be highly relevant to the chemistry that occurs in astronomical environments, they are not performed under interstellar cloud conditions, and hence, the same reaction pathways may not be operative. In this context, the authors should carefully review passages in their manuscript where this connection is made, emphasizing if specific types of reactions might be viable in space, i.e. are the likely to have a barrier to reaction.

Accept with minor revision.

Reviewer #2 (Remarks to the Author):

This paper presents an experimental study on the formation of PAHs in a naphthalene:Ar electrical discharge with the goal of elucidating the formation of PAHs in the interstellar medium. Fragments, products and reaction intermediates are structurally identified by mass selective IR and UV laser spectroscopy. The experiments provide evidence for the formation of larger PAHs containing up to four cyclic rings in the gas phase. In addition, the experiments provide evidence for resonance stabilized radical (RSR) intermediates and intermediates that contain di-acetylenic side groups, alongside di-acetylene molecules. A reaction mechanism is put forward in which di-acetylene plays a

key role in the growth of large PAHs and this is presented as an alternative to the generally accepted HACA mechanism for PAH growth in sooting environments.

The experimental method uses a molecular beam to cool down the reaction products and ion-dip spectroscopy to identify abundant species. This is a novel approach and the authors should be commended for this. However, I see three major issues with this paper as it is and cannot accept it for publication.

1) No attempt is made to link the conditions in the experiment to those in the interstellar medium. There is a vague reference to the detection of di-acetylene in the source CRL 618 as an argument that di-acetylene is an important interstellar molecule. Actually, these observations pertain to the outflow from a star not to the interstellar medium. Di-acetylene is not expected to be an abundant interstellar molecule. Calculated abundances in molecular clouds are at the $1E-9$ level. So, collision time scales are ~ 100 Myr. That makes it not very relevant. The other vague reference supporting astrophysical relevance is that plasma sources have been used to create radicals of astrophysical interest (c.f., p3). However, these studies were done as a convenient way to produce relevant species and study their spectroscopic signatures as the basis for a search in astronomical spectra. These studies did not pretend to mimic interstellar conditions. To be fair, other studies have been published in reputable journals – including Nature – where the focus is on the reaction mechanism that could be of interest to astrochemistry. However, the relevance of the experiments to ISM conditions has to be addressed. The core should be to identify relevant species in the experiments and make plausible that these species are relevant in space as well and the experimental conditions are relevant to space. Perhaps, the goal of this paper is to highlight the possible role of diacetylene in the growth of large PAHs in plasma's and astronomy is taken as a convenient vehicle to promote the wider interest. I don't see that as necessary. The reaction mechanism underlying PAH and soot formation itself has a wide impact.

2) The authors seem to hint that the reactions take place in the expanding molecular beam. However, no actual evidence for this is presented. More likely, the reactions take place in the discharge zone where reactive radicals are abundant rather than in the cold, collision free molecular beam. The paper does not discuss the conditions in the reaction zone. There is an extensive literature on PAH discharges – including naphthalene ones and the statement "The growth mechanisms of larger PAHs have not yet been explored in such plasma environments" is not accurate. Let me point the authors to Allati et al., J. Phys. Chem. A 2019, 123, 2107-2113 as an entry

point to the literature. This study identifies the breakdown products of naphthalene in a plasma ($C_{10}H_7^+$ and $C_{10}H_6^+$ but also C_4H_2 and C_2H_2). They also point out the importance of acetylene in the growth of larger species. The presence of mass 152 and 176 in their mass spectra is taken as evidence for the HACA mechanism with acetylene. The paper by Contreras and Salama *Astrophys. J., Suppl. Ser.* 2013, 208, 6 is of interest as well. The present paper adds to these studies by using ion-dip spectrometry to structurally characterize the species involved rather than just relying on mass spectrometry. Similar studies have been done in sooting flames but then using laser induced fluorescence. I am not aware of an ion-dip study under such conditions and that is well worth highlighting. Nevertheless, the authors should quantify the conditions of their experiments and relate their results to other studies and other techniques. As the potential role of diacetylene in the growth is a key point of this paper, this has to be more firmly established. That a stable species is abundant may just be because it is unreactive. In the proposed reaction mechanism (figure 4), it is indeed the C_4H radical that drives the growth (although I could imagine that electron excitation in the plasma produces metastable excited diacetylene which would be more reactive). In the HACA mechanism, the PAH is "radicalized" through H abstraction followed by reaction with acetylene. If diacetylene is the key, still H abstraction would play a central role. All of this drives the study away from astrophysical relevant conditions and that may be where the impact of this paper should be: ion-dip spectroscopy as a tool to identify reaction products under plasma or flame conditions and the putative identification of a new reaction mechanism.

3) Ion-dip spectroscopy is the new tool in this paper and these experiments are state of the art. The supporting quantum chemistry is not, though. The authors clearly point out the failure of double harmonic DFT calculations to help identify the absorbing species. This is in terms of clear offsets in peak position as well as a failure to reproduce key combination bands. It is not clear to me why the authors do not use Gaussian16 to calculate anharmonic spectra. Those should result in much better agreement with experimental spectra, require no scaling factor and can predict frequency as well as intensity of combination bands. The authors go through a song and a dance to argue this issue away based upon good agreement between experiments and reference spectra. This is an excellent point but why hide this away in figure S7 in the supplements? Indeed, why compare to unreliable quantum chemistry calculations at all. It is very hard to evaluate the "goodness" of the fits and the derived mixing ratios from the results presented in fig 2 & 3 or S2, S3, S4, and S5. The authors should make the comparison with reference spectra the central point. I appreciate that this requires a concerted effort to measure relevant spectra but then the conclusions are robust. And their elegant probing technique is done full right.

Reviewer #1 (Remarks to the Author):

The manuscript by Lemmens et al. is a well-written and well-reasoned article that describes the spectroscopic analysis using mass-selective IR and UV techniques when naphthalene is subjected to an electrical discharge. Theoretical calculations are used to aid the identification of various fragments and larger species; I found this portion of the analysis fairly convincing. Key findings include the prominence of intermediate species with C₄H side chains and larger PAHs with 3 and even 4 membered rings. Although the present findings may provide insight into the pathways by which larger PAH molecules are synthesized, it would have been helpful if the authors had demonstrated the same basic findings for a second small PAH. Perhaps for obvious reasons, this experiment might be beyond the scope of the present work, but when and where appropriate, the authors should discuss the generality of their present results. Is there any reason to believe that other small PAHs might behave differently?

We thank the reviewer for her/his positive summary and remarks. The generality of the work can indeed be demonstrated by applying the same technique to other PAHs such as phenanthrene or fluorene. We believe that presenting this data is beyond the scope of this article, since this work gives a complete picture of the strength of the method in a short communication. It is a first showcase of the IR ion dip technique applied to unambiguously assign the structures in addition to the mass spectrometry in molecular beam discharges of PAHs. Nevertheless, you provide a very interesting point and in fact we are currently conducting/have conducted such experiments in our laboratory indeed showing the wide applicability of the method.

It is also worth emphasizing that although the present experiments may indeed be highly relevant to the chemistry that occurs in astronomical environments, they are not performed under interstellar cloud conditions, and hence, the same reaction pathways may not be operative. In this context, the authors should carefully review passages in their manuscript where this connection is made, emphasizing if specific types of reactions might be viable in space, i.e. are the likely to have a barrier to reaction.

We agree with your point on the direct connection between laboratory conditions and astronomical environments. We are aware that the discharge does not operate under interstellar cloud conditions (densities are much higher in our experiment). We have addressed this issue throughout the paper (highlighted in yellow). Note also that the electron energy in circumstellar regions is of the same order of magnitude as the determined electron energy in our plasma¹⁻³.

Accept with minor revision.

Reviewer #2 (Remarks to the Author):

We thank the reviewer for her/his comments with respect to our manuscript. We have divided her/his comments in smaller sections in order to provide a point-by-point answer. In general we agree with the suggestions and questions, and some of the remarks were already presented, but maybe hidden in the manuscript. We have rephrased some parts for clarity.

This paper presents an experimental study on the formation of PAHs in a naphthalene:Ar electrical discharge with the goal of elucidating the formation of PAHs in the interstellar medium. Fragments, products and reaction intermediates are structurally identified by mass selective IR and UV laser spectroscopy. The experiments provide evidence for the formation of larger PAHs containing up to four cyclic rings in the gas phase. In addition, the experiments provide evidence for resonance stabilized radical (RSR) intermediates and intermediates that contain di-acetylenic side groups, alongside di-acetylene molecules. A reaction mechanism is put forward in which di-acetylene plays a key role in the growth of large PAHs and this is presented as an alternative to the generally accepted HACA mechanism for PAH growth in sooting environments.

The experimental method uses a molecular beam to cool down the reaction products and ion-dip spectroscopy to identify abundant species. This is a novel approach and the authors should be commended for this. However, I see three major issues with this paper as it is and cannot accept it for publication.

We would like to thank the reviewer for the acknowledgement of our novel approach to identify and assign reaction products by mass selective IR spectroscopy in molecular beam discharges. In the following we address the issues and improve and clarify the paper where necessary.

1) No attempt is made to link the conditions in the experiment to those in the interstellar medium. There is a vague reference to the detection of di-acetylene in the source CRL 618 as an argument that di-acetylene is an important interstellar molecule. Actually, these observations pertain to the outflow from a star not to the interstellar medium. Di-acetylene is not expected to be an abundant interstellar molecule. Calculated abundances in molecular clouds are at the $1E-9$ level. So, collision time scales are ~ 100 Myr. That makes it not very relevant. The other vague reference supporting astrophysical relevance is that plasma sources have been used to create radicals of astrophysical interest (c.f., p3). However, these studies were done as a convenient way to produce relevant species and study their spectroscopic signatures as the basis for a search in astronomical spectra. These studies did not pretend to mimic interstellar conditions. To be fair, other studies have been published in reputable journals – including Nature – where the focus is on the reaction mechanism that could be of interest to astrochemistry. However, the relevance of the experiments to ISM conditions has to be addressed. The core should be to identify relevant species in the experiments and make plausible that these species are relevant in space as well and the experimental conditions are relevant to space. Perhaps, the goal of this paper is to highlight the possible role of diacetylene in the growth of large PAHs in plasma's and astronomy is taken as a convenient vehicle to promote the wider interest. I don't see that as necessary. The reaction mechanism underlying PAH and soot formation itself has a wide impact.

We do agree with your point that the aim of this article is to identify relevant species in the experiments using mass selective IR spectroscopy as novel tool and to make plausible that these species are also relevant in astrochemistry. We are aware that the discharge does not operate under interstellar cloud conditions (densities are much higher in our experiment). We have addressed this issue throughout the paper and referred to literature that discusses the role of low-energy free electrons in astrochemistry (highlighted in yellow). The astronomical relevance of the experimental conditions can be found in the electron density, i.e. the electron energy in circumstellar regions is of

the same order of magnitude as the determined electron energy in our plasma¹⁻³. We have addressed your concerns by adding the following to the text:

“In stellar outflows or the ISM, energetic particles and photons that ionize atoms and molecules create free low-energy electrons^{2,3}. In these ionized gas regions, the low-energy electrons can induce gas phase chemistry. Much is still unclear on the growth mechanisms of PAHs in such plasma environments, mainly because one has had to rely solely on mass spectrometry⁴⁻⁷. We create relevant conditions by an electrical discharge. The energy of the electrons in our experiment is estimated between 3.15 and 3.45 eV of which two would be needed to break a C=C bond. The molecular density is higher in our experiment than in the relevant astrophysical objects in order to produce the necessary amount of reaction products.”

Besides this point, we would like to point out that the study is of astrochemical relevance as a result of the specific molecule that is investigated. Gas phase reaction mechanisms under before unknown conditions will aid in elucidating the full formation pathways of PAHs.

2) The authors seem to hint that the reactions take place in the expanding molecular beam. However, no actual evidence for this is presented. More likely, the reactions take place in the discharge zone where reactive radicals are abundant rather than in the cold, collision free molecular beam.

We fully agree with your remark. This was also what we tried to report. For clarity we have rephrased the following. We replaced the sentence:

“an isolating spacer of 6 mm is used to confine the plasma directly after discharge in order to increase the number of collisions before expansion thereby allowing for longer reaction time.”

By

“an isolating spacer of 6 mm is used to confine the plasma directly after discharge in order to increase the number of collisions before the expansion to create a longer reaction time. The subsequent expansion “freezes” the reaction after which the reaction mixture is probed.”

See also chapter 3 of the supplementary information.

The paper does not discuss the conditions in the reaction zone.

In our manuscript, we aimed to describe the most relevant reaction conditions, which we have extended and clarified. We do give an estimate of what we consider the most important parameter in our experiment, the electron energy. To address your concerns more we have also added an estimate of the (partial) pressure in our discharge zone.

We replaced:

“Using the Paschen curve (for γ between 0.1 and 0.2) and the relation between electron energy and pressure from Engel, the electron energy is estimated to be 3.15-3.45 eV”

By

“Using the Paschen curve (for γ between 0.1 and 0.2⁸ which corresponds to a pressure between 0.37 and 0.51 mbar) and the relation between electron energy and pressure from Engel⁹, the electron energy is estimated to be 3.15-3.45 eV.”

We have also added:

“The molecular densities is higher in our experiment than in most relevant astrophysical objects in order to produce the necessary amount of reaction products.”

At last, from the partial pressure an estimate number density of naphthalene is derived as highlighted in yellow in the paper.

There is an extensive literature on PAH discharges – including naphthalene ones and the statement “The growth mechanisms of larger PAHs have not yet been explored in such plasma environments” is not accurate. Let me point the authors to Allati et al., J. Phys. Chem. A 2019, 123, 2107-2113 as an entry point to the literature. This study identifies the breakdown products of naphthalene in a plasma (C₁₀H₇⁺ and C₁₀H₆⁺ but also C₄H₂ and C₂H₂). They also point out the importance of acetylene in the growth of larger species. The presence of mass 152 and 176 in their mass spectra is take as evidence for the HACA mechanism with acetylene.

The paper by Contreras and Salama Astrophys. J., Suppl. Ser. 2013, 208, 6 is of interest as well. The present paper adds to these studies by using ion-dip spectrometry to structurally characterize the species involved rather than just relying on mass spectrometry. Similar studies have been done in sooting flames but then using laser induced fluorescence. I am not aware of an ion-dip study under such conditions and that is well worth highlighting.

We thank the reviewer for pointing out this interesting literature. We were not aware of this very recent study on the plasma chemistry of naphthalene. Of course we have included a reference. Nevertheless, we still think that our work is of interest since our technique does not only provide a mass but the addition of mass selective IR spectroscopy uniquely allows us to assign the observed masses to multiple (“exotic”) molecular structures unambiguously. The paper by Contreras and Salama Astrophys. J., Suppl. Ser. 2013, 208, 6 is indeed of interest since it brings forward a similar astrochemically relevant discharge source, but also lacks the unambiguous assignment of reaction products.

Nevertheless, the authors should quantify the conditions of their experiments and relate their results to other studies and other techniques.

Please refer back to our reply on your remark: “The paper does not discuss the conditions in the reaction zone.” Where we address your remark on the quantification of the conditions in our experiment. The relation of our technique to other discharge nozzles is difficult since most studies do not supply the pressure in the discharge region. We initiate the possibility of relating these studies by determining a pressure, number density and electron energy. The latter is carefully determined by measuring the breakdown voltage and plasma current (not the set-values of a power supply).

As the potential role of diacetylene in the growth is a key point of this paper, this has to be more firmly established. That a stable species is abundant may just be because it is unreactive. In the proposed reaction mechanism (figure 4), it is indeed the C₄H radical that drives the growth (although I could imagine that electron excitation in the plasma produces metastable excited diacetylene which would be more reactive). In the HACA mechanism, the PAH is “radicalized” through H abstraction followed by reaction with acetylene. If diacetylene is the key, still H abstraction would play a central role. All of this drives the study away from astrophysical relevant conditions and that

may be where the impact of this paper should be: ion-dip spectroscopy as a tool to identify reaction products under plasma or flame conditions and the putative identification of a new reaction mechanism.

As the reviewer highlights, the aim of this paper is indeed: IR ion-dip spectroscopy as a tool to unambiguously identify reaction products under plasma conditions. Following the suggestions by the reviewer, we have stressed this more in the abstract of the article.

Moreover, with the identified products we are able to suggest a reaction mechanism involving diacetylene radical as key ingredient. If the HACA mechanism would play a central role this would be associated with an increase of the m/z 127 of the radicalized naphthalene. In our m/z vs. current scans we do not observe a significant increase in this mass channel. These results suggest the importance of other mechanisms present, as was previously suggested as well by McCarthy and Leone^{10,11} or now in our paper, where hydrogen abstraction is not the starting point but rather the “final” step in the reaction.

3) Ion-dip spectroscopy is the new tool in this paper and these experiments are state of the art. The supporting quantum chemistry is not, though. The authors clearly point out the failure of double harmonic DFT calculations to help identify the absorbing species. This is in terms of clear offsets in peak position as well as a failure to reproduce key combination bands. It is not clear to me why the authors do not use Gaussian16 to calculate anharmonic spectra. Those should result in much better agreement with experimental spectra, require no scaling factor and can predict frequency as well as intensity of combination bands.

Thank you for your suggestion. We are aware of the possibilities of Gaussian16 to perform anharmonic frequency calculations as we for example also did in our previous work on PAHs¹². We have investigated the potential diagnostic value of the anharmonic calculations. As you can see in the figure below, we did perform calculations using the “anharmonic” option in gaussian (example of 2-ethynyl naphthalene). However, although having used several basis sets, we cannot get reasonable results for most molecules, i.e. a general reliable “anharmonic” method is not yet found.

The fundamental frequencies and intensities are drastically different and therefore not only the diagnostic value of those but also of the overtones or combination bands is lost. The “anharmonicity” of these molecules is an interesting problem, but beyond the scope of our work here. We think that the harmonic calculations suffice for identification, with the knowledge that the 1210 cm^{-1} peak originates from a $\equiv\text{CH}$ group for which we use the reference spectra as substantiation.

The authors go through a song and a dance to argue this issue away based upon good agreement between experiments and reference spectra. This is an excellent point but why hide this away in figure S7 in the supplements? Indeed, why compare to unreliable quantum chemistry calculations at all. It is very hard to evaluate the “goodness” of the fits and the derived mixing ratios from the results presented in fig 2 & 3 or S2, S3, S4, and S5. The authors should make the comparison with reference spectra the central point. I appreciate that this requires a concerted effort to measure relevant spectra but then the conclusions are robust. And their elegant probing technique is done full right.

Assignment based on reference spectra does not have our preference based on the following arguments:

1. Quantum chemical calculations are sufficiently reliable to make assignments as quantified in our current and previous work¹². It is still the most direct way to obtain structural information from IR spectra.
2. Assignment of molecular structures to IR spectra is based on a trial and error method where sometimes more than 20 trial structures are calculated and compared. These calculated IR spectra are highly discriminating, and a match with the experiment is often very clear. For many of the molecules that we have identified, no reference compounds are readily available and more importantly even unstable.

We therefore deliberately chose to uniformly assign structures using calculations. However, we have validated this method with reference spectra or in the cases where our approach demands extra experiments (for example the 1210 cm^{-1} band), reference spectra are obtained and presented in the supplementary information.

References:

- (1) Mason, N. J.; Nair, B.; Jheeta, S.; Szymańska, E. Electron Induced Chemistry: A New Frontier in Astrochemistry. *Faraday Discuss.* **2014**, *168*, 235–247. <https://doi.org/10.1039/c4fd00004h>.
- (2) Indriolo, N.; McCall, B. J. Cosmic-Ray Astrochemistry. *Chem. Soc. Rev.* **2013**, *42* (19), 7763–7773. <https://doi.org/10.1039/c3cs60087d>.

- (3) Tielens, A. G. G. M.; Meixner, M. M.; van der Werf, P. P.; Bregman, J.; Tauber, J. A.; Stutzki, J.; Rank, D. Anatomy of the Photodissociation Region in the Orion Bar. *Science* (80-.). **1993**, *262*, 86–89. <https://doi.org/10.1126/science.262.5130.86>.
- (4) Alliat, M.; Donaghy, D.; Tu, X.; Bradley, J. W. Ionic Species in a Naphthalene Plasma: Understanding Fragmentation Patterns and Growth of PAHs. *J. Phys. Chem. A* **2019**, *123* (10), 2107–2113. <https://doi.org/10.1021/acs.jpca.9b00100>.
- (5) Hobrock, D. L.; Kiser, R. W. Electron Impact Studies of Trihalomethanes. *J. Phys. Chem.* **1964**, *68* (3), 575–579.
- (6) Gillon, X.; Houssiau, L. Plasma Polymerization Chemistry of Unsaturated Hydrocarbons: Neutral Species Identification by Mass Spectrometry. *Plasma Sources Sci. Technol.* **2014**, *23* (4). <https://doi.org/10.1088/0963-0252/23/4/045010>.
- (7) West, B.; Joblin, C.; Blanchet, V.; Bodi, A.; Sztáray, B.; Mayer, P. M. On the Dissociation of the Naphthalene Radical Cation: New IPEPICO and Tandem Mass Spectrometry Results. *J. Phys. Chem. A* **2012**, *116* (45), 10999–11007. <https://doi.org/10.1021/jp3091705>.
- (8) Lieberman, M. A.; Lichtenberg, A. J. *Principles of Plasma Discharges and Materials Processing*; John Wiley & Sons, Inc.: Hoboken, NJ, USA, 2005.
- (9) Von Engel, A. *Ionized Gases*; American Institute of Physics: New York, 1994.
- (10) McGuire, B. A.; Burkhardt, A. M.; Kalenskii, S.; Shingledecker, C. N.; Remijan, A. J.; Herbst, E.; McCarthy, M. C. Detection of the Aromatic Molecule Benzonitrile (c-C₆H₅CN) in the Interstellar Medium. *Science* **2018**, *359*, 202–205.
- (11) Trevitt, A. J.; Goulay, F.; Taatjes, C. A.; Osborn, D. L.; Leone, S. R. Reactions of the CN Radical with Benzene and Toluene: Product Detection and Low-Temperature Kinetics. *J. Phys. Chem. A* **2010**, *114* (4), 1749–1755. <https://doi.org/10.1021/jp909633a>.
- (12) Lemmens, A. K.; Rap, D. B.; Thunnissen, J. M. M.; Mackie, C. J.; Candian, A.; Tielens, A. G. G. M.; Rijs, A. M.; Buma, W. J. Anharmonicity in the Mid-Infrared Spectra of Polycyclic Aromatic Hydrocarbons. Molecular Beam Spectroscopy and Calculations. *Astron. Astrophys.* **2019**, *130*, 1–10. <https://doi.org/10.1051/0004-6361/201935631>.

Reviewers' comments:

Reviewer #1 (Remarks to the Author):

The revised manuscript appears to meaningfully address the feedback of the two reviewers, and in doing so improves the clarity of the paper.

Accept as is.

Reviewer #2 (Remarks to the Author):

A key point of this study is that reaction products are identified through an IR-UV ion-dip method. That is exciting, opening up new avenues to study the reactions involved in PAH growth at a deeper level. That this method can truly do so has to be well established in the narrative. I am therefore deeply puzzled by the statement that the quantum chemical calculation suffice for identification purposes in the analysis of the IR spectra when clearly major bands are not reproduced in figure 2 and in addition frequency shifts are very apparent. To spell it out: The calculated pattern of some bands resembles the measured spectra but the main band to be included in the analysis is the 1200 cm^{-1} band and that one is not. The authors do discuss this and refer to the SI. But the decomposition in panels b,c,d,e in figure 2 are not convincing. Let me just reiterate, the spectra have to be compared to standards and indeed the comparison in Figure S7 is the most convincing part in this analysis. The spectroscopy community is aware that shifts and missing bands occur more often in such comparisons. However, the authors should consider that their paper will be read by an interested but lay public – astronomers – that will not be able to judge this properly and may summarily dismiss their study.

In the 2nd paragraph on p3, the authors try to place their experimental conditions into an astrophysical context. That discussion is misguided though. While cosmic rays or UV photons create electrons with energies of a few eV, these electrons quickly lose their energy through interaction with other electrons and the most abundant atoms and molecules (H, H₂). This interaction heats the gas but does not play much of a role in the chemistry. The astronomical audience will be well aware of this and might disregard the paper because of it. I do think this paper is relevant for astrochemistry; not as an analogue study but rather as one that identifies new growth routes for PAHs. Perhaps, it is time to retreat here and to accept – and spell out – that the conditions in the

plasma do not resemble those in space. Rather, the paper could highlight that radical-neutral reactions involving diacetylenic derivatives can be important for the growth of PAHs. The last sentence of the abstract catches this well.

We would like to thank both reviewers for their positive feedback on our revised manuscript discussing the use of mass-selective IR-UV spectroscopy for the characterization of PAH formation after discharge. We especially would like to acknowledge the efforts of reviewer 2 to help us to make our manuscript understandable for a broad community as well. We discuss his/her concerns and remarks in detail below, and we have highlighted significant changes in yellow in our manuscript.

Comments from reviewer 2:

A key point of this study is that reaction products are identified through an IR-UV ion-dip method. That is exciting, opening up new avenues to study the reactions involved in PAH growth at a deeper level. That this method can truly do so has to be well established in the narrative. I am therefore deeply puzzled by the statement that the quantum chemical calculation suffice for identification purposes in the analysis of the IR spectra when clearly major bands are not reproduced in figure 2 and in addition frequency shifts are very apparent. To spell it out: The calculated pattern of some bands resembles the measured spectra but the main band to be included in the analysis is the 1200 cm^{-1} band and that one is not. The authors do discuss this and refer to the SI. But the decomposition in panels b,c,d,e in figure 2 are not convincing. Let me just reiterate, the spectra have to be compared to standards and indeed the comparison in Figure S7 is the most convincing part in this analysis. The spectroscopy community is aware that shifts and missing bands occur more often in such comparisons. However, the authors should consider that their paper will be read by an interested but lay public – astronomers – that will not be able to judge this properly and may summarily dismiss their study.

Indeed this manuscript presents a novel, orthogonal fashion of characterizing PAH formation and growth which goes beyond mass analysis alone, since measuring only the mass would give you the chemical composition but not the chemical structure. By combining mass-to-charge detection with IR and UV spectroscopy, we are able to obtain structural information. For retrieving information on the chemical structure, we either rely on reference spectra (if available or possible to measure) and quantum chemical calculations (which widely applicable for all species; transient, short-lived, neutral, charged or radical). This latter, more general approach -the assignment method using DFT calculations- is a well-established method for structure elucidation in several disciplines ranging from identification of peptides¹⁻⁵, metabolites⁶, reaction intermediates^{7,8}, clusters^{9,10} and also PAHs^{11,12}. This approach works well for unbranched PAHs as can be seen in Figure 3 of the manuscript and in literature^{13,14}. However, when possible we have added reference spectra, as for example can be seen for pyrene which is measured under jet-cooled conditions in Fig. 3c to exemplify this in the manuscript.

When PAHs are formed with an ethynyl side-group, the experimental spectra show an additional band at 1210 cm^{-1} . We agree with the reviewer that the analysis of this 1210 cm^{-1} band deserves more attention and we have now included anharmonic computations. We have clarified this by an additional section on page 7. First of all, we have used this 1210 cm^{-1} peak to identify that the formed species is an ethynyl-substituted PAH. This peak appeared for all ethynyl-substituted PAHs, as we observe in our work here, but also in previous studies, a key conclusion is that this peak acts as a diagnostic tool for the $\equiv\text{CH}$ functional group. We have confirmed this by obtaining reference spectra as well as references in literature and textbook¹⁵⁻¹⁹.

Regarding these anharmonic calculations, as we aimed to elucidate in our previous response, in anharmonic calculations the fundamental bands are shifted by a large and undesired extent. Following the advice from the scientists of the quantum chemical program package, we have performed new calculations, where we have removed the derivatives for these miscalculated modes in the anharmonic analysis and included these calculations for the assigned structures in the article. In these anharmonic analysis the 1210 cm⁻¹ band is indeed present.

We agree that confirming our assignment based on the agreement between our experiments and theory would be ideal. Unfortunately recording reference spectra of all compounds is not possible as some are instable species and some synthesis routes are not reasonably achievable. For that reason, we combine here the DFT calculated IR spectra and mass-selected IR spectra to assign structures, and have confirmed our assignment by reference spectra where possible. These two reference spectra are now included in figure 2 in the main article. In addition, besides for 1- and 2-ethynyl naphthalene reference spectra are added for molecules including 1-vinylnaphthalene, 2-vinylnaphthalene, pyrene, acenaphthylene, 1,4-diethynylbenzene that all show excellent agreement with theory (see supplementary information).

In addition to above changes, a more detailed illustration and description of the assignment is given in the supplementary information.

In the 2nd paragraph on p3, the authors try to place their experimental conditions into an astrophysical context. That discussion is misguided though. While cosmic rays or UV photons create electrons with energies of a few eV, these electrons quickly lose their energy through interaction with other electrons and the most abundant atoms and molecules (H, H₂). This interaction heats the gas but does not play much of a role in the chemistry. The astronomical audience will be well aware of this and might disregard the paper because of it. I do think this paper is relevant for astrochemistry; not as an analogue study but rather as one that identifies new growth routes for PAHs. Perhaps, it is time to retreat here and to accept – and spell out – that the conditions in the plasma do not resemble those in space. Rather, the paper could highlight that radical-neutral reactions involving diacetylenic derivatives can be important for the growth of PAHs. The last sentence of the abstract catches this well.

We agree with the reviewer on this point. To focus on the strengths of our approach, we have rewritten the 2nd paragraph on p3 to:

“In stellar outflows or the ISM energetic particles and photons ionize atoms and molecules, thereby inducing gas-phase synthesis of larger molecules^{20,21}. In our experiment, we perform ionization/radicalization and subsequently induce gas-phase reactions by means of an electrical discharge. **Although conditions are not analogues**, reaction routes that occur in this discharge are of interest for astrochemistry as newly identified growth pathways for PAHs. Much is still unclear on the growth mechanisms of PAHs in these plasma environments, mainly because one has had to rely solely on mass spectrometry^{22–25}. Here, we identify reaction products and intermediates through mass selective IR-UV ion dip spectroscopy and study the reactions of PAH growth at a deeper level.”

We have also rephrased several other instances of “suitable controlled conditions” to “controlled conditions”.

References used in this point-by-point report:

- (1) Martens, J.; Grzetic, J.; Berden, G.; Oomens, J. Structural Identification of Electron Transfer

- Dissociation Products in Mass Spectrometry Using Infrared Ion Spectroscopy. *Nat. Commun.* **2016**, *7*, 11754.
- (2) Kamrath, M. Z.; Garand, E.; Jordan, P. A.; Leavitt, C. M.; Wolk, A. B.; Stipdonk, M. J. Van; Miller, S. J.; Johnson, M. A. Vibrational Characterization of Simple Peptides Using Cryogenic Infrared Photodissociation of H₂-Tagged Mass-Selected Ions. *J. Am. Chem. Soc.* **2011**, *133*, 6440–6448.
 - (3) Wu, R.; McMahon, T. B. Infrared Multiple Photon Dissociation Spectra of Proline and Glycine Proton-Bound Homodimers. Evidence for Zwitterionic Structure. *J. Am. Chem. Soc.* **2007**, *129*, 4864–4865.
 - (4) Seo, J.; Warnke, S.; Pagel, K.; Bowers, M. T.; Helden, G. Von. Infrared Spectrum and Structure of the Homochiral Serine Octamer-Dichloride Complex. *Nat. Chem.* **2017**, *9* (12), 1263–1268.
 - (5) Bakels, S.; Porskamp, S. B. A.; Rijs, A. M. Peptide Aggregation Formation of Neutral Peptide Aggregates as Studied by Mass-Selective IR Action Spectroscopy *Angewandte. Angew. Chemie - Int. Ed.* **2019**, *58*, 10537–10541.
 - (6) Martens, J.; Outersterp, R. E. Van; Vreeken, R. J.; Cuyckens, F.; Coene, K. L. M.; Engelke, U. F.; Kluijtmans, L. A. J.; Wevers, R. A.; Buydens, L. M. C.; Redlich, B.; et al. Infrared Ion Spectroscopy : New Opportunities for Small-Molecule Identification in Mass Spectrometry - A Tutorial Perspective. *Anal. Chim. Acta* **2019**. <https://doi.org/10.1016/j.aca.2019.10.043>.
 - (7) Chiavarino, B.; Crestoni, M. E.; Fornarini, S.; Lanucara, F.; Lemaire, J.; Maître, P. Meisenheimer Complexes Positively Characterized as Stable Intermediates in the Gas Phase. *Angew. Chemie - Int. Ed.* **2007**, *46*, 1995–1998.
 - (8) Roithova, J. Characterization of Reaction Intermediates by Ion Spectroscopy. *Chem Soc Rev* **2012**, *41*, 547–559.
 - (9) Gruene, P.; Rayner, D. M.; Redlich, B.; Meer, A. F. G. Van Der; Lyon, J. T.; Meijer, G.; Fielicke, A. Structures of Neutral Au₇, Au₁₉, and Au₂₀ Clusters in the Gas Phase. *Science* **2008**, *321*, 674–677.
 - (10) Gruenloh, C. J.; Carney, J. R.; Arrington, C. A.; Zwier, T. S.; Fredericks, S. Y.; Jordan, K. D. Infrared Spectrum of a Molecular Ice Cube: The S₄ and D_{2d} Water Octamers in Benzene-(Water)₈. *Science* **1997**, *276*, 1678–1681.
 - (11) Lorenz, U. J.; Solcà, N.; Lemaire, J.; Maître, P.; Dopfer, O. Infrared Spectra of Isolated Protonated Polycyclic Aromatic Hydrocarbons : Protonated Naphthalene. *Angew. Chemie - Int. Ed.* **2007**, *46*, 6714–6716.
 - (12) Bouwman, J.; De Haas, A. J.; Oomens, J. Spectroscopic Evidence for the Formation of Pentalene⁺ in the Dissociative Ionization of Naphthalene. *Chem. Commun.* **2016**, *52* (12), 2636–2638.
 - (13) Lemmens, A. K.; Rap, D. B.; Thunnissen, J. M. M.; Mackie, C. J.; Candian, A.; Tielens, A. G. G. M.; Rijs, A. M.; Buma, W. J. Anharmonicity in the Mid-Infrared Spectra of Polycyclic Aromatic Hydrocarbons. Molecular Beam Spectroscopy and Calculations. *Astron. Astrophys.* **2019**, *628* (A130), 1–10.
 - (14) Maltseva, E.; Mackie, C. J.; Candian, A.; Petrigiani, A.; Huang, X.; Lee, T. J.; Tielens, A. G. G. M.; Oomens, J.; Buma, W. J. High-Resolution IR Absorption Spectroscopy of Polycyclic Aromatic Hydrocarbons in the 3 Mm Region: Role of Hydrogenation and Alkylation. *Astron. Astrophys.* **2018**, *610* (A65).

- (15) Constantinidis, P.; Hirsch, F.; Fischer, I.; Dey, A.; Rijs, A. M. Products of the Propargyl Self-Reaction at High Temperatures Investigated by IR/UV Ion Dip Spectroscopy. *J. Phys. Chem. A* **2017**, *121* (1), 181–191.
- (16) Bellamy, L. J. *The Infra-Red Spectra of Complex Molecules*; Chapman and Hall: London, 1975.
- (17) Nyquist, R. A.; Potts, W. J. Infrared Absorptions Characteristic of the Terminal Acetylenic Group ($-C\equiv C-H$). **1960**, *16*, 419–427.
- (18) Evans, J. C.; Nyquist, R. A. The Vibrational Spectra and Vibrational Assignments of the Propargyl Halides. *Spectrochim. Acta* **1963**, *19*, 1153–1163.
- (19) Wallace, W. E. *Phenylethyne IR Spectrum, NIST Chemistry WebBook, NIST Standard Reference Database Number 69*; P.J. Linstrom and W.G. Mallard, Ed.; National Institute of Standards and Technology: Gaithersburg MD, 2019.
- (20) Indriolo, N.; Mc Call, B. J. Cosmic-Ray Astrochemistry. *Chem. Soc. Rev.* **2013**, *42* (19), 7763–7773.
- (21) Tielens, A. G. G. M.; Meixner, M. M.; van der Werf, P. P.; Bregman, J.; Tauber, J. A.; Stutzki, J.; Rank, D. Anatomy of the Photodissociation Region in the Orion Bar. *Science* **1993**, *262*, 86–89.
- (22) Allati, M.; Donaghy, D.; Tu, X.; Bradley, J. W. Ionic Species in a Naphthalene Plasma: Understanding Fragmentation Patterns and Growth of PAHs. *J. Phys. Chem. A* **2019**, *123* (10), 2107–2113.
- (23) Hobrock, D. L.; Kiser, R. W. Electron Impact Studies of Trihalomethanes. *J. Phys. Chem.* **1964**, *68* (3), 575–579.
- (24) Gillon, X.; Houssiau, L. Plasma Polymerization Chemistry of Unsaturated Hydrocarbons: Neutral Species Identification by Mass Spectrometry. *Plasma Sources Sci. Technol.* **2014**, *23* (4).
- (25) West, B.; Joblin, C.; Blanchet, V.; Bodi, A.; Sztáray, B.; Mayer, P. M. On the Dissociation of the Naphthalene Radical Cation: New IPEPICO and Tandem Mass Spectrometry Results. *J. Phys. Chem. A* **2012**, *116* (45), 10999–11007.

REVIEWERS' COMMENTS:

Reviewer #2 (Remarks to the Author):

I am satisfied with the revisions.